# A Transcriptome and Methylome Study Comparing Tissues of Early and Late Onset Colorectal Carcinoma

**DOI:** 10.3390/ijms232214261

**Published:** 2022-11-17

**Authors:** Muhammad G Kibriya, Maruf Raza, Anthony Quinn, Mohammed Kamal, Habibul Ahsan, Farzana Jasmine

**Affiliations:** 1Institute for Population and Precision Health (IPPH), Biological Sciences Division, The University of Chicago, Chicago, IL 60637, USA; 2Department of Pathology, Jahurul Islam Medical College, Kishoregonj 2336, Bangladesh; 3Department of Pathology, The Laboratory Dhaka, Dhaka 1205, Bangladesh

**Keywords:** colorectal carcinoma, early onset CRC, multi-omics, lymphovascular invasion, perineural invasion, DNA damage repair, DNA replication repair, MMR, base excision repair, FASN, *CTLA4*, *HAVCR2*

## Abstract

There is an increase in the incidence of early onset colorectal carcinoma (EOCRC). To better understand if there is any difference in molecular pathogenesis of EOCRC and late onset colorectal carcinoma (LOCRC), we compared the clinical, histological, transcriptome, and methylome profile of paired CRC and healthy colonic tissue from 67 EOCRC and 98 LOCRC patients. The frequency of stage 3 CRC, lymph node involvement, lymphovascular invasion, and perineural invasion was higher in the EOCRC group. Many of the cancer related pathways were differentially expressed in CRC tissue in both EOCRC and LOCRC patients. However, the magnitude of differential expression for some groups of genes, such as DNA damage repair genes and replication stress genes, were significantly less pronounced in the EOCRC group, suggesting less efficient DNA damage repair to be associated with EOCRC. A more marked methylation of “growth factor receptor” genes in LOCRC correlated with a more pronounced down-regulation of those genes in that group. From a therapeutic point of view, more over-expression of fatty acid synthase (*FASN*) among the LOCRC patients may suggest a better response of *FASN* targeted therapy in that group. The age of onset of CRC did not appear to modify the response of cis-platin or certain immune checkpoint inhibitors. We found some differences in the molecular pathogenesis in EOCRC and LOCRC that may have some biological and therapeutic significance.

## 1. Introduction

Incidence of colorectal cancer (CRC) among young adults has increased over the last 25 years [1]. In a study involving a large cohort of 143.7 million people aged 20–49 years from 20 European countries, a total of 187,918 individuals (0.13%) were diagnosed with CRC. From 2004 to 2016, on average, CRC incidence increased by 7.9% per year among subjects aged 20–29 years, 4.9% per year among subjects aged 30–39 years, and 1.6% per year in the age group of 40–49 years [1]. Considering the current practice of a preventive colonoscopy recommendation at 50 years of age, some considered CRC before the age of 50 years as early onset CRC (EOCRC) [1,2,3,4,5]. Other studies have used 40 years as a cut-off for EOCRC and late onset CRC (LOCRC) [6,7,8]. There are some interesting findings, such as the association of EOCRC and obesity [9,10], the cause of which is undetermined. In a previous study, we also reported a high proportion of CRC patients in Bangladesh presenting at or below the age of 40 years [11]. There is a growing interest in exploring the underlying causes of EOCRC. Like some other studies [6,7,8], we divided the cases as EOCRC and LOCRC as =<40 years and >40 years, respectively.

The clinical features of EOCRC may be different from those of LOCRC. Some studies suggested that EOCRC is usually detected in the distal colon and rectum, whereas in the older age group it is in the proximal colon [3,12]. The anatomical location of EOCRC may provide important insights into the underlying disease processes and treatment responses, since embryologically the proximal and distal colon are different. EOCRC display different histopathological features when compared to late onset cases. Poor differentiation, perineural invasion (PNI), venous invasion, and mucinous and/or signet cell morphology, all of which are suggestive of an unfavorable tumor biology and associated with worse oncological outcomes, are more common among patients with EOCRC [6,8]. It was also found that EOCRC patients present with higher rates of metastasis and recurrence compared to their older counterparts. Regarding the outcome of EOCRC and LOCRC, some studies reported a worse prognosis for EOCRC, while others found equivalent or superior outcomes among the younger patients [8,13,14,15,16].

A study showed that, among EOCRC, approximately 30% of patients were affected by tumors harboring mutations causing hereditary cancer predisposing syndromes, and 20% have familial CRC [17]. Others also found this incidence among 19.7% to 29.4% of EOCRC, depending on the age of onset [7,18]. Chang et al. found that, among EOCRC, 17% demonstrated abnormalities in DNA mismatch repair and 5% had known germline genetic disorders [6]. Therefore, the remaining 50% of cases were of sporadic EOCRC. The rate of EOCRC is increasing, and it shows some distinct characteristics regarding the location, anatomical features, and pathological features. There is a growing interest to explore the findings at the molecular level, which can potentially lead to improved treatment and survival.

It is largely unclear if there are any differences in the transcriptome or methylome profile among the EOCRC and LOCRC patients that can (a) shed light on the difference in the molecular pathogenesis in these two groups and (b) provide a molecular basis of potential use of different treatment options. To address these research questions, in this study, we compared the clinical, histological, transcriptome, and methylome profiles of paired CRC and surrounding healthy colonic tissue from the same patients who presented with CRC at =<40 years of age and those who presented after 40 years of age.

## 2. Results

In this collection of 165 CRC patients from Bangladesh, we included all consecutive patients during the multiple time periods spanning between December 2009 and May 2016, irrespective of the age of onset of CRC. Of the 165 patients, 67 of them (40% of the patients) were of age =< 40 years. The comparison of clinical and histological features among the patients with LOCRC (age > 40 years) and EOCRC (age =< 40 years) is shown in Table 1. We did not find any statistically significant difference for sex, tumor location, tumor grading, presence of tumor infiltrating leukocyte (TIL) status, presence of signet ring appearance, CEA level, microsatellite instability (MSI), KRAS mutation, BRAF mutation, EGFR mutation, and relative telomere length (RTL) shortening. It may be noted that the frequency of BRAF mutation was low in this population, and we did not find an EGFR mutation in any patient. The analysis showed that the EOCRC patients more frequently had advanced stage CRC and lymph node involvement than LOCRC patients (see Table 1). In fact, we also found that the EOCRC patients more frequently had a lympho-vascular invasion (LVI) and PNI than the LOCRC patients did (see Table 1).

Histological staging has a strong association with LVI (13.5%, 23.8%, and 50% in stage 1, stage 2, and stage 3, respectively, *p* = 0.0001, Chi square test) and PNI (2.7%, 7.1%, and 20.9% in stage 1, stage 2, and stage 3, respectively, *p* = 0.01, Chi-square test). Therefore, we looked at the association between the age of onset and LVI and PNI by stratifying the data by stage (see Table 2). Compared to LOCRC, EOCRC more frequently had LVI only in the presence of the stage 2 disease and more frequently had PNI only in the presence of stage 3 disease. In other words, these associations of LVI and EOCRC or PNI and EOCRC were not independent of staging.

### 2.1. Differential Gene Expression in CRC

#### 2.1.1. Age of Onset of CRC and Differential Gene Expression of Cancer Related Gene Sets

In the next step, we asked if the differential expression (CRC tissue vs. paired normal colonic tissue) of some known cancer related gene sets were different in magnitude in EOCRC and LOCRC patients. The detailed list of these gene sets is shown in Appendix A. Table 3 shows that the “Growth factor receptor” genes were significantly down-regulated in tumor tissue only in LOCRC [fold change −1.12 (95% CI: from −1.16 to −1.08%), *p* = 1.21 × 10^−8^], but not in EOCRC patients (see Figure 1). Tumor suppressor genes (TSG) were significantly down-regulated in both LOCRC and EOCRC patients by a similar magnitude. However, anti-TSG (cyclin D group of genes) were significantly up-regulated, both in LOCRC and EOCRC patients, but the magnitude of over-expression was significantly more pronounced in LOCRC than in EOCRC patients [fold change 1.46 (95% CI 1.38–1.55) vs. 1.29 (95% CI 1.20–1.39), ANOVA interaction *p* = 0.007] (see Figure 2). DNA repair genes as a whole were significantly over-expressed in both groups, but to a lesser extent in the young onset group. Therefore, in the next step, we tested the gene sets involved in different DNA repair mechanisms separately.

#### 2.1.2. Age of Onset of CRC and Differential Gene Expression of DNA Damage Repair Gene Sets

The detailed gene list is shown in Appendix A. Except for genes for translesion synthesis (TLS), genes involved in all other DNA repair mechanisms were significantly over-expressed in CRC tissue, irrespective of the age of onset (see Table 4). However, in comparison to the magnitude of over-expression in LOCRC, the over-expression of genes related to “mismatch repair” (see Appendix A), “Fanconi anemia” (see Figure 3), “non-homologous end joining (NHEJ)”(see Figure 4), and “microhomology mediated end joining (MMED)” was less pronounced in EOCRC patients (see Table 4). This may suggest a potentially low efficiency of the DNA damage repair mechanism in EOCRC.

#### 2.1.3. Age of Onset of CRC and Differential Expression of Replication Stress Gene-Sets

In the light of the above results for the association of the age of onset of CRC and DNA damage repair machinery, we explored if the genes involved in different biological processes in replication stress are different depending on the age of onset of CRC. The total list of genes at the replication stress site and their functional group [19,20] that we tested are shown in Appendix A. Analyses of our data (see Table 5) suggested that, in response to the cancer, on average, the genes related to “DNA replication repair” were over-expressed in CRC tissue, irrespective of the age of onset. However, it was significantly less pronounced in EOCRC patients compared to LOCRC patients [fold change 1.09 (95%CI 1.08–1.10) vs. 1.14 (95%CI 1.13–1.15), ANOVA interaction *p* = 2.54 × 10^−8^] (see Appendix A). In the same line, the metabolism related genes were also over-expressed in both, but less pronounced in EOCRC (see Appendix A)

#### 2.1.4. Age of Onset of CRC and Differential Expression of Notch4-GATA4-IRG1 axis Gene-Sets

A recent study suggested the possible association of “NOTCH4-GATA4-IRG1 axis” genes and “leptin and other obesity related genes” with CRC [21,22]. The gene list is shown in Appendix A. We tested that in our data (see Figure 5). We found that the “Notch4-GATA-IRG” set of genes were over-expressed in tumor tissue by the same magnitude in LOCRC [fold change 1.09 (95% CI 1.07–1.12)] and in EOCRC [fold change 1.08 (95% CI 1.06–1.12)] (ANOVA interaction *p* = 0.71).

We observed the down-regulation of *LEPR* and *GHRL* in CRC, but it was not different by the age of onset (see Appendix A). However, we found that the fatty acid synthase (*FASN*) gene was significantly over-expressed in LOCRC [fold change 1.69 (95%CI 1.41–2.03)] compared to a non-significant increase in EOCRC patients [1.25-fold (95% CI: −1.009 to 1.59)] (see Appendix A). The over-expression of *FASN* has been reported in CRC [23,24]. The data may suggest *FASN* as a therapeutic target in the LOCRC group, as has been suggested by others [25,26].

#### 2.1.5. Age of Onset of CRC and Differential Expression of KEGG Pathways

In addition to examining the selective biologically relevant gene sets, as shown above, we also tested all of the KEGG pathways to see if the magnitude of the differential expression (CRC vs. normal colon tissue) of any pathway was different in LOCRC and EOCRC patients. The detailed result is shown in Appendix A. It was interesting to see that, among all of the KEGG pathways, the most significant interaction was seen in the “DNA replication pathway”. Genes in the “DNA replication pathway” were over-expressed by 1.27-fold (95%CI 1.25–1.28) in LOCRC patients compared to 1.16-fold (95% CI 1.14–1.18) in EOCRC patients (ANOVA interaction *p* = 2.73 × 10^−15^).

#### 2.1.6. Association of Differentially Expressed “Gene Sets” with Histology in CRC

Considering the significance of the tumor stage, LVI and PNI, we tried to see if the gene expression of some of the top most differentially expressed gene sets (CRC vs. normal tissue, EORC and LOCRC combined) were different in magnitude by these histological changes. Assuming that the cancer cells respond to the DNA damage by increasing the DNA damage repair and DNA replication machinery, one can expect that these gene sets will be up-regulated in CRC tissue compared to normal colon tissue. Our data suggests that these cellular responses for repairing the DNA damage are lower in magnitude in the presence of LVI, PNI, and advanced stage. For example, in the absence of LVI, the base excision repair (BER) genes are on average 1.13-fold (95% CI 1.11–1.14) over-expressed, but in the presence of LVI, it is 1.07-fold (95% CI 1.05–1.08). The details are shown in Table 6, which shows that perhaps LVI had the strongest influence on the magnitude of differential expression for these DNA repair machineries. A photomicrograph example of LVI and PNI is shown in Appendix A, respectively.

#### 2.1.7. Gene Expression Profiling from Therapeutic Point of View

**T-cell inflamed gene expression profile (GEP)**: This group of genes has been used to predict immunotherapy targeting programmed cell death protein-1 (PD-1, also known as CD274)) [27,28]. We looked at the gene expression of these genes in our patients to see if they were different by age of onset (see Figure 6A,B). In both EOCRC and LOCRC groups, these genes were down-regulated to a similar extent. However, stratification by MSI status showed that the down-regulation was more pronounced (ANOVA interaction *p* = 1.7 × 10^−7^) in the MSS group [fold change −1.32 (95% CI from −1.28 to −1.36)] than in the MSI group [fold change −1.19 (95% CI from −1.13 to −1.25)].

**Other ICI target genes**: We also looked at targets of some other immune check-point inhibitors—*LAG3, PDCD1, CD274, CTLA4,* and *HAVCR2.* Again, the age of onset did not influence the differential expression (interaction *p* = 0.17) (see Figure 6C,D), but the MSI status influenced the overall differential expression, as we reported in an earlier publication [11]. *LAG3, PDCD1,* and *CD274 (PDL-1)* were down-regulated in CRC (see Figure 6E,F), but *CTLA4* and *HAVCR2* were up-regulated only in the MSI–CRC group, suggesting a potential benefit from *CTLA4* and *HAVCR2* inhibitors in that subset only [11] (see Figure 6F).

**Platinum drug resistance genes:** The differential expression was similar in magnitude in EOCRC and LOCRC. The MSI status also did not have any influence either (see Figure 6G,H). The overall slight down-regulation of these genes in CRC suggested a low potential chance of development of cis-platin resistance.

In summary, on the basis of the gene expression profile of genes relevant to therapy (a), it may not be surprising if *PD1* blockers do not show promising results in this CRC population, but some other ICI, such as *CTLA4* or *HAVCR2* blockers, may have promising results in the MSI subgroup; (b) the possibility of platinum drug resistance may be low, irrespective of age of onset in this population.

### 2.2. DNA Methylation

#### Age of Onset of CRC and Differential DNA Methylation

First, we carried out the differential methylation analysis (paired CRC tissue vs. corresponding healthy colon tissue) at the individual probe level data. Using paired analysis, we found a large number of differentially methylated loci (DML), both in LOCRC (n = 143,646, at FDR 0.05) and in EOCRC (n = 111,066, at FDR 0.05) (see Figure 7A). A large number of these DML (n = 100,370, which is 69.8% of those found in LOCRC and 90.36% of those found in EOCRC) were common between the two groups (see Figure 7A). These lists of DML do not take the magnitude of the differential methylation (delta beta = beta value of CRC tissue—beta value of normal) into account. Considering our aim to detect the DML, the magnitudes of delta beta of which are significantly different among LOCRC and EOCRC, we used an ANOVA model that included an interaction term “tissue x age of onset of CRC”. Our analysis suggested that there were 5607 loci with interaction *p* =< 0.05. These three lists were used in the Venn diagram (Figure 7B). The intersections of the Venn diagram allowed us to identify the following groups of DML:I.DML common in both EOCRC and LOCRC and the magnitude of delta beta was not different (n = 97,939) as the interaction *p* was >=0.05. This represents the largest group of DML in CRC. The scatterplot in Figure 7C shows the magnitude of delta beta of these DML in LOCRC and EOCRC and the cut-off lines on both axes show a large number of loci with delta beta > 0.1 or <−0.1, indicating hyper- or hypo-methylation exceeding 10%.II.DML common in both EOCRC and LOCRC, but the magnitude of delta beta was significantly different between EOCRC and LOCRC patients (n = 2431).III.DML found only in EOCRC, but the magnitude of the delta beta was not different between the EOCRC and LOCRC (n = 9925). The scatterplot in Figure 7D shows that the magnitudes of delta beta of these DML were low and not different among LOCRC and EOCRC.IV.DML specific to EOCRC (n = 771): these DML are differentially methylated only in EOCRC, and the magnitude of the delta beta is significantly more pronounced from that seen in LOCRC. The details of these DML are presented in Appendix A. The scatterplot in Figure 7E shows the magnitudes of delta beta of these DML; only a few exceeded 0.1 or 10% differential methylation. Regardless of the low magnitudes of delta beta of these 771 EOCRC specific DML, because of specificity, these markers were able to separate CRC from normal among the EOCRC patients (see the PCA plot in Figure 7F). The only hypermethylated DML with delta beta >= 0.1 was the *SOX8* gene. Among the three hypomethylated loci with delta beta =< 0.1, one was the *TACC1* gene (see Figure 7E), which is known to be associated with other cancers. The methylation status of the *TACC1* gene in LOCRC and EOCRC is shown in Figure 7G,H, respectively, showing that *TACC1* was hypomethylated only in EOCRC, but not in LOCRC.V.DML specific to old age onset CRC (1946): these DML are differentially methylated only in LOCRC, and the magnitude of the delta beta is significantly more pronounced from that seen in EOCRC. The details of these DML are presented in Appendix A. For many of these DML, the magnitudes of delta beta exceeded 0.1 or 10% differential methylation.

In the next step, we tried to identify if the magnitude of the differential methylation of some gene set(s) was different among the patients with LOCRC compared to those with EOCRC patients. We included an interaction term “tissue x age of onset” in the gene set ANOVA model(s); the *p*-value of that interaction term identified pathways/gene sets that were differentially methylated depending on the age of onset.

Among the cancer related gene sets, the genes involved in “Growth factor receptors” and “DNA repair” showed different magnitudes of differential methylation among EOCRC and LOCRC (see Appendix A). This methylation data for “Growth factor receptor” genes also correlate to our gene expression data in a sense that, for these genes, on average, we observed the hypermethylation of DNA in CRC tissue in general (more in LOCRC than EOCRC) and the down-regulation in mRNA in CRC tissue (more pronounced in LORC than in EOCRC). Among the DNA damage related gene sets, “Direct reversal Repair” genes were differentially methylated depending on the age of onset (see Appendix A). Among the replication stress site gene sets, “Immune regulation” genes were differentially methylated depending on the age of onset (see Appendix A). Finally, the top KEGG pathways which showed different magnitudes of differential methylation among EOCRC and LOCRC were “Antigen processing and presentation”, “Type I diabetes mellitus”, “JAK-STAT signaling pathway” etc. (see Appendix A).

In summary, (a) the vast majority of the DML in CRC were common in EOCRC and LOCRC; (b) the vast majority of the robust DML [the one with the magnitude of delta beta => 0.2 (20%)] were common between EOCRC and LOCRC; (c) although the EOCRC specific DML were relatively small in number and also small in the magnitude of delta beta. These EOCRC specific markers could potentially differentiate cancer from healthy tissue.

## 3. Discussion

The incidence of CRC among young adults has increased globally [1,29,30,31]. Siegel RL et al. published a comprehensive overview of current CRC statistics in the United States, including the estimated numbers of new cases and deaths in 2020 by age and incidence, survival, and mortality rates and trends by age and race/ethnicity based on incidence data through 2016 and mortality data through 2017 [31].

In this study from a Bangladeshi population, we attempted to find any histological and transcriptome wide and methylome wide differences in CRC tissue compared to corresponding normal tissue among EOCRC and LOCRC patients. We acknowledge that such a study cannot shed light on the etiology of the increase in incidence in EOCRC. Additionally, we did not have any post-surgical follow-up data to comment on the difference in prognosis of EOCRC, if any. From the histology perspective, we documented that EOCRC is associated with an advanced stage. This association has been seen in other populations as well. However, the greater proportion of patients with advanced stage could not be simply explained by a delay in diagnosis. The other two histological features associated with EOCRC were LVI and PNI. The significance of KRAS and BRAF mutations in CRC is well known [32]. However, we did not see any difference in the frequency of these mutations between EOCRC and LOCRC in our series.

The histological diagnosis of LVI on examination of the hematoxylin & eosin (H&E)-stained slide includes the presence of tumor cells within a vascular space; erythrocytes surrounding the tumor cells; the identification of endothelial cells lining the space; the presence of an elastic lamina surrounding the tumor; and the attachment of tumor cells to the vascular wall [33]. PNI is a pathologic process characterized by a tumor invasion of nervous structures and spread along nerve sheaths. The pathogenesis of PNI likely involves complex signaling between tumor cells, stromal cells, and the nerves. PNI is known to be a marker for a more aggressive tumor phenotype and poor prognosis in several malignancies, most notably head and neck and prostate cancers. PNI was defined as tumor cells within any layer of the nerve sheath or tumor in the perineural space that involved at least one third of the nerve circumference [34]. To our knowledge, our study is the first to show the association of these histological markers of advanced disease (stage, LVI, and PNI) with an underlying transcriptomic profile of impairment in DNA damage repair machinery in CRC tissue.

In one retrospective study among LOCRC, 207 right sided and 207 left sided tumors are compared after curative resection, the authors found that the left-sided tumor exhibited better survival outcomes than the right-sided ones after curative resection [4]. Survival data for EOCRC is conflicting. Survival rates among the young and older groups also show variation depending on the CRC stages. The right sided tumor exhibited a more advanced stage, increased tumor size, more frequently poorly differentiated tumors, more harvested lymph nodes, and more positivity of LVI than left sided ones. They also found better 5-year survival outcomes for the group with left sided CRC [4]. Right sided CRC presents more with iron deficiency anemia [35]. Powell et al. compared 134 (33%) right sided, 125 (30%) left sided, and 152 (37%) rectal tumors. Emergency presentation (*p* < 0.001), anemia (*p* < 0.001), advanced stage (*p* < 0.001), poor differentiation (*p* < 0.001), and older age (*p* < 0.05) were more commonly observed in right sided cancers [36].

There are some differences in the clinicopathological characteristics among EOCRC and LOCRC. The differences between the 94 EOCRC (avg. 27 years) and 275 LOCRC (avg. 67 years) were studied [37]. There were differences in the stage at diagnosis—stage III and IV (76% vs. 46%), signet ring-cell (13% vs. 1%), poorly differentiated (37% vs. 8%), MSI status (27% vs. 13%), and the 5-year disease specific survival (48% vs. 78%) in EOCRC and LOCRC, respectively [37], and no difference was found in KRAS and BRAF mutations. Low BRAF mutations were reported among Bangladeshi immigrants to the UK [38]. Another study also found that, in sporadic CRC, independent of the age, the LVI and PNI were associated with a poor prognosis and recurrence [39]. In our study, we found advanced stage, LVI, and PNI to be significantly associated with EOCRC compared to LOCRC.

Very few studies have compared the gene expression difference between the EOCRC and LOCRC. Tunca B et al. looked at the expression profiles of 114 different genes which were evaluated using mRNA PCR arrays in 39 tumors and 20 surgical margin tissue samples from 39 sporadic CRC patients diagnosed at less than 50 years of age [40]. The expression levels of *IMPDH2*, *CK20*, *MAP3K8*, and *EIF5A* were strongly up-regulated in CRC tissues compared with normal colorectal tissues, but not compared to the LOCRC. A similar study was carried out on sporadic EOCRC and healthy controls (<50 years old). Seven genes, *CYR61*, *UCHL1*, *FOS*, *FOS B*, *EGR1*, *VIP*, and *KRT24*, were consistently up-regulated in the mucosa of all six patients compared with the mucosa from four healthy controls [41].

Gene expression variation between FFPE tissues from six EOCC patients (<50 years) and six LOCRC patients (>65 years) were examined. Among the 770 genes assayed, changes in expression levels of 88 genes were unique (28 up- and 60 down-regulated) to EOCC (using the cutoff criteria of expression level differences > 2-fold and *p* value < 0.01) [42]. At the pathway level, *RAS*, *MAPK*, *WNT*, and DNA repair pathways were similarly deregulated in both age groups, whereas *PI3K*-AKT signaling was more specific to EOCC and cell cycle pathways to LOCC [42]. Berg M et al. performed an integrated analysis of copy number changes and gene expression in 23 sporadic EOCRC patients with a median age of 44 years and 17 LOCRC patients with median age of 79 years [43]. Tissues were preserved in buffer RLT. In the younger group, *CLC*, *LTBP4,* and *ZNF57*4 were up-regulated and *PPAT, RG9MTD2*, *EIF4E,* and *PLA2G12A* were down-regulated compared to the older group. Agesen TH et al. compared gene expression in EOCRC and LOCRC and found that a good number of genes are over- or under-expressed in EOCRC [44]. Among those, Charcot–Leyden crystal protein (CLC) was 10 times over-expressed and interferon (alpha, beta, and omega) receptor-1 (IFNAR1) was under-expressed.

Gene expression data in EOCRC were analyzed by Mo X et al. [45]. A total of 140 module hub genes were identified and found to be enriched in the ‘mitochondrial large ribosomal subunit’, ‘structural constituent of ribosome’, ‘poly (A) RNA binding’, ‘collagen binding’, ‘protein ubiquitination’, and ‘ribosome pathway’. Twenty-six module hub genes were found to have a degree score > 5 in the PPI network, seven of which [secreted protein acidic and cysteine rich (*SPARC*), decorin (*DCN*), fibrillin 1 (*FBN1*), WW domain containing transcription regulator 1 (*WWTR1*), transgelin (*TAGLN*), DEAD-box helicase 28 (*DDX28*), and cold shock domain containing C2 (*CSDC2*)], had good prognostic values for patients with early-onset CRC, but not late-onset CRC. Therefore, the authors suggested that *SPARC*, *DCN*, *FBN1*, *WWTR1*, *TAGLN*, *DDX28,* and *CSDC2* may contribute to the development of early-onset CRC and may serve as potential diagnostic biomarkers.

Structurally, leptin has similarity with other proteins of the cytokine family, belonging to the group of cytokines commonly called adipocytokines or adipokines. Initially described as an anti-obesity hormone, leptin has subsequently been shown to also influence hematopoiesis, thermogenesis, reproduction, angiogenesis, and immune homeostasis [46]. Some variants of the Leptin gene have been found to be associated with CRC in females [9]. There is an inverse correlation between adiponectin and leptin in obesity [47]. In our study, we observed the down-regulation of LEPR GHRL in both EOCRC and LOCRC without a significant difference; however *FASN* was significantly over-expressed in LOCRC compared to a non-significant increase in EOCRC patients.

In a recent paper, Joo JE et al. compared the DNA methylation of tumor and healthy mucosa in 110 EOCRC (<50 years), 334 intermediate onset CRC (IOCRC) (50–70 years), and 325 LOCRC (>70 years) [48]. They used FFPE samples on Methylation 450K chips using the DNA restoration kit. They found extensive DNA methylation alterations in all CRCs, including EOCRCs. They identified DNA methylation-related changes specific to EOCRC, including *TFAP2A* and *GSX1* genes, and 12 differentially methylated genes associated with the MODY pathway. In our study, we found a relatively smaller number of EOCRC specific DML, but those could cluster the CRC samples well from normal tissue. Some of our methylation findings also correlated with gene expression findings in EOCRC.

We did not have any CRC patients with distal metastasis in this series and we did not have chemotherapy or immunotherapy data. However, we tried to look for the molecular basis in the CRC tissue for the potential use of ICI in these EOCRC and LOCRC patients. The age of onset did not appear to be a factor that can modify the choice of ICI. Rather, the MSI status can be used to select a potential subset of CRC patients for certain ICI. With all of the limitations of our study in mind, some of the strengths of the study may be noted. First, paired tumor-normal samples from the same individual for comparison is the most robust method for detecting any gene expression and methylation changes in cancer. Second, we used tissue samples preserved in RNA later for RNA and fresh frozen samples for DNA, which are the gold standard for such assays. Third, to our knowledge, this is one of the first studies from native Bangladeshi patients with CRC to comprehensively see a difference between EOCRC and LOCRC pathogenesis from the molecular perspective.

## 4. Materials and Methods

For this study, we included 330 paired (tumor and adjacent normal) samples from 165 CRC patients (m = 96, f = 69). Of them, 33 had right-sided CRC (cecum 8, ascending colon 17, hepatic flexure 4, and transverse colon 4) and 132 had left-sided CRC (descending colon 8, sigmoid colon 15, recto-sigmoid junction 7, and rectum 102). Sixty-seven of the patients (39 male and 28 female) were aged =<40 years. The fresh frozen samples were collected from 165 CRC patients from the department of Pathology, Bangabandhu Sheikh Mujib Medical University (BSMMU), Dhaka, Bangladesh at different times spanning between December 2009 and May 2016. The patients were at different stages of CRC (stage-1: 37, stage 2: 42 stage 3: 86). From each patient, the specimens were collected from the surgically resected tumor and the surrounding unaffected part of the colon about 5–10 cm away from the tumor mass. A surgical pathology fellow collected all samples from the operating room immediately after the surgical resection. A histopathology examination was performed on H&E stained slides in routinely processed paraffin impregnated tissue blocks. The slides were examined independently by two pathologists and there was concordance in all 165 cases. For the staging and grading of the CRC, the World Health Organization Classification of tumors was followed [49]. From each individual, we obtained a pair of tumor and normal tissues, which were frozen immediately and shipped on dry ice to the molecular genomics lab at The University of Chicago for subsequent DNA, RNA extraction, and molecular assay.

For each patient, we also abstracted key demographic and clinical data and tumor characteristics from hospital medical records. Written informed consent was obtained from all participants. The research protocol was approved by the “Ethical Review Committee, Bangabandhu Sheikh Mujib Medical University”, Dhaka, Bangladesh (BSMMU/2010/10096), and by the “Biological Sciences Division, University of Chicago Hospital Institutional Review Board”, Chicago, IL, USA (10-264-E).

### 4.1. DNA and RNA Extraction and Quality Control

DNA was extracted from fresh frozen tissue using a Puregene Core kit (Qiagen, MD, USA). The electropherogram from Agilent BioAnalyzer with Agilent DNA 12,000 chips showed the fragment size to be >10,000 bp. RNA was extracted from RNA Later preserved colonic tissue using a Ribopure tissue kit (Ambion, Austin, TX, USA, Cat# AM1924).

### 4.2. Relative Telomere Length (RTL) Measurement

For RTL measurement, we used a Luminex-based assay using QuantiGene Plex chemistry (Invitrogen, Santa Clara, CA, USA). The details of the assay are described earlier [50,51]. Briefly, the assay requires ~50 ng of DNA, which is hybridized to sequence-specific probes for the telomere repeat sequence (TEL) and reference gene sequence (*ALK)*. The TEL and *ALK* gene signals are amplified using branched DNA technology and detected using Luminex technology. We used custom designed probes to measure the abundance of the telomere repeat sequence. The 24-mer probe targeted four repeats–“TTAGGGTTAGGGTTAGGGTTAGGG”. As a reference single gene, we used *ALK* which showed very stable copy numbers (CN = 2) in all the DNA samples detected by oligonucleotide-based microarray SNP chips from our previous study. The result is a ratio, and hence there is no unit. The assay precision was good to excellent, with an intra-class correlation coefficient (ICC) of 0.91 (95% CI 0.86–0.94) [50]. The RTL assay failed in 17 samples (nine CRC and eight normal tissue) out of the total 330 samples tested. This failure rate (5.1%) was similar to what we have seen previously in a larger scale study using the same Luminex-based RTL measurement assay [52]. RTL data of this series of CRC patients was recently published [53].

### 4.3. Genome-Wide Gene Expression Assay

We used microarray data (Illumina HT12 v4 BeadChip) from the first 71 paired tumor and normal tissue RNA (of the same set of 165 patients used for RTL assay in this study). The chip contains a total of 47,231 probes covering 31,335 genes. Therefore, for many genes, there were multiple probes on the chip targeting different genomic regions of the same gene, that also enables the covering of multiple isoforms, if any. We used the probe level data. Paired samples were processed in the same chip (12 samples/chip). One sample from the normal tissue failed on the microarray. Therefore, we had gene expression data from 71 CRC tissues and 70 corresponding normal tissues. Gene expression data was normalized using quantile normalization in the GenomeStudio software.

### 4.4. Genome-Wide Methylation Assay

We also have methylation data (Illumina HumanMethylation450 DNA analysis BeadChip v1.0 Assay) from the first 125 paired tumor-normal samples (125 pairs out of the same set of 165 patients used for this study) [11]. The DNA samples were subjected to bisulfite conversion using EZ-96 DNA Methylation Kit (Zymo Research, Irvine, CA, USA). The chip presents 485,577 loci of which there are 150,254 in CpG Island, 112,067 in Shore (0–2 kb from island), 47,114 in Shelf (2–4 kb from the island), and 176,112 in deep sea (>4 kb from CpG island). We did not include the markers in the deep sea region in the final differential methylation analysis. Paired samples (CRC and corresponding normal) were processed on the same chip to avoid a batch effect. From this assay, on average 17 loci per gene were interrogated. A Tecan Evo robot was used for automated sample processing and the chips were scanned on a single iScan reader. If the intensity of the methylated loci is X and the intensity of the unmethylated loci is Y, then, the methylation score (beta value) is X/X + Y. If all are unmethylated (X = 0), then the methylation level is 0/0 + Y = 0. If all loci are methylated (Y = 0), then the beta value is X/X + 0 = 1. If 50% of probes are hybridized at the methylated loci and 50% are hybridized at the unmethylated loci, then the methylation score is 50/50 + 50 = 0.5.

### 4.5. Microsatellite Instability (MSI) Detection

A high-resolution melting (HRM) analysis method was used for the detection of two mononucleotide MSI markers—BAT25 and BAT26 [54,55]. A tumor was defined as having MSI when it showed instability with at least one of these markers (BAT25 and BAT26), and as MSS when it showed no instability for both of the markers. We used published primer sequences [54]. The amplification conditions included the polymerase activation step at 95 °C for 2 min, followed by five cycles of denaturation at 95 °C for 15 s, annealing starting at 60 °C for 30 s, extension at 72 °C for 30 s, and an additional 33 cycles of denaturation at 95 °C for 15 s, annealing at 53 °C for 30 s, and extension at 72 °C for 30 s. Before the HRM step, the products were heated to 95 °C for 1 min and cooled to 40 °C for 1 min, to allow for the heteroduplex formation. HRM was carried out and the data was collected over the range from 60 to 95 °C, with temperature increments of 0.2 °C/seat each 0.05 s. The BAT25 and BAT26 products were sequenced for validation. In this way, a total of 30 tumor samples showed MSI and all were confirmed by another relatively novel MSI marker CAT25 [56,57].

### 4.6. KRAS and BRAF Mutation Detection:

Tumor and adjacent healthy colonic tissue from 165 paired (tumor and normal) tissues were tested for KRAS (rs 112445441) and BRAFV600E mutations by high resolution melt analysis, as described previously [55].

### 4.7. Statistical Analysis

To compare the continuous variables, we used a *t*-test or one-way analysis of variance (ANOVA). The principal component analysis (PCA) and sample histograms were checked as a part of quality control analyses of the microarray data. Mixed-model multi-way ANOVA (which allows more than one ANOVA factor to be entered in each model) was used to compare the individual probe level expression data (for gene expression) or the beta value of CpG loci (for methylation data) across different groups. For statistical analysis, we used Partek Genomics Suite (version 7.0) (https://www.partek.com/partek-genomics-suite/, accessed on 14 November 2022). In general, “tissue” (tumor/adjacent normal), age of onset of CRC (0: =<40 years, 1: >40 years), LVI (0 = no, 1 = yes), PNI (0 = no, 1 = yes), telomere shortening (0 = no, 1 = yes), and MSI status (MSI/MSS) were used as categorical variables with fixed effect. These levels represent all conditions of interest, whereas “person ID#” (as proxy of inter-person variation) was treated as a categorical variable with random effect, since the person ID is only a random sample of all of the levels of that factor. The method of moments estimation was used to obtain estimates of variance components for mixed models [58]. As per the study design, we processed both the CRC tissue and the corresponding adjacent normal sample from one individual in a single chip. In the ANOVA model, the log_2_-transformed gene expression or beta-value for the CpG loci were used as the response variable (Y), and “Tumor” (tumor or normal), person ID#, “MSI-status”, and “age of onset” were entered as ANOVA factors.

For the paired analysis, we used the following model:$$Y_{ijk}=\mu +Tumor_{i}+Person_{j}+\varepsilon_{ijk}$$
where Y*ijk* represents the *k-th* observation on the *i-th* tumor *j-th* person. *Μ* is the common effect for the whole experiment. *Εijk* represents the random error present in the *k-th* observation on the *i-th* tumor *j-th* person. The errors *εijk* are assumed to be normally and independently distributed with mean 0 and standard deviation *δ* for all measurements. Person is a random effect.

For the detection of the interaction between tumor and age of onset, the following model was used:$$Y_{ijk}=\mu +Tumor_{i}+Age\;of\;onset\;of\;CRC_{J}+Tumor*Age\;of\;onset\;of\;CRC_{ij}+\varepsilon_{ijk}$$
where Y*ijk* represents the *k-th* observation on the *i-th* tumor *j-th* age of onset. *Μ* is the common effect for the whole experiment. *Εijk* represents the random error present in the *k-th* observation on the *i-th* tumor *j-th* age of onset of CRC. The errors *εijk* are assumed to be normally and independently distributed with mean 0 and standard deviation *δ* for all measurements.

Gene ontology (GO) was used to group a set of genes into a category. In GO Enrichment analysis, we tested if the genes found to be differentially expressed or methylated fell into a gene ontology category more often than expected by chance [59]. We used a chi-square test to compare the “number of significant genes from a given category/total number of significant genes” vs. “number of genes on chip in that category/total number of genes on the microarray chip”. The negative log of the p-value for this test was used as the enrichment score. Therefore, a GO group with a high enrichment score represents a lead functional group. The enrichment scores were analyzed in a hierarchical visualization and in tabular form.

Gene set ANOVA is a mixed model ANOVA to test the expression or methylation of a set of genes (sharing the same category or functional group) instead of an individual gene in different groups (https://www.partek.com/partek-genomics-suite/, accessed on 14 November 2022). The analysis is performed at the gene level, but the result is expressed at the level of the gene set-category by averaging the member genes’ results. The equation for the model was:Model: Y = μ + T + P + G + S (T ∗ P) + ε
where Y represents the expression or methylation status of a gene set category, μ is the common effect or average expression/methylation of the gene set category, T is the tissue-to-tissue (tumor/normal) effect, P is the patient-to-patient effect, G is the gene-to-gene effect (differential expression or methylation of genes within the gene set category independent of tissue types), S (T*P) is the sample-to-sample effect (this is a random effect, and nested in the tissue and patient), and ε represents the random error. All of the figures were generated using Partek Genomics Suite (version 7.0) (https://www.partek.com/partek-genomics-suite/, accessed on 14 November 2022).

## 5. Conclusions

We found that a high proportion of CRC patients presented at =<40 years of age. Histologically, EOCRC was more frequently associated with advanced stage, LVI, and PNI. As for molecular mechanisms for pathogenesis, the genome-wide gene expression and methylation suggested that the EOCRC was associated with an impaired DNA damage repair response, DNA replication repair, and immune response. From a therapeutic perspective, the study suggested the potential use of *FASN* targeted therapy, especially in the LOCRC group in this population.

## Figures and Tables

**Figure 1 ijms-23-14261-f001:**
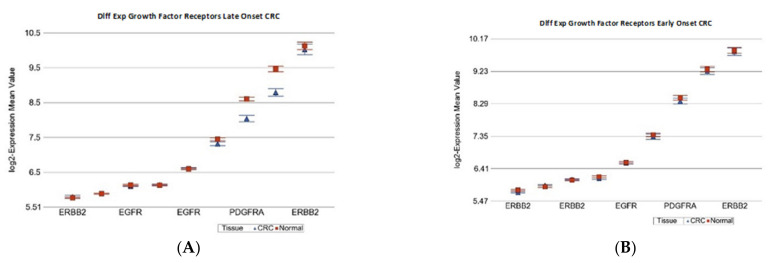
Differential gene expression of growth factor receptor genes in paired CRC tissue (in blue) and healthy colonic mucosa (in red). Gene probes are arranged on the *x*-axis by expression level, and the mean of log_2_ transformed expression value is shown on the *y*-axis. For many genes, there were multiple probes on the chip. Gene symbols for all of the gene probes could not be shown on the *x*-axis. Data from patients from the LOCRC group are shown on the left (**A**) showing down-regulation; and data from EOCRC group are shown on the right (**B**), where there was no differential expression found.

**Figure 2 ijms-23-14261-f002:**
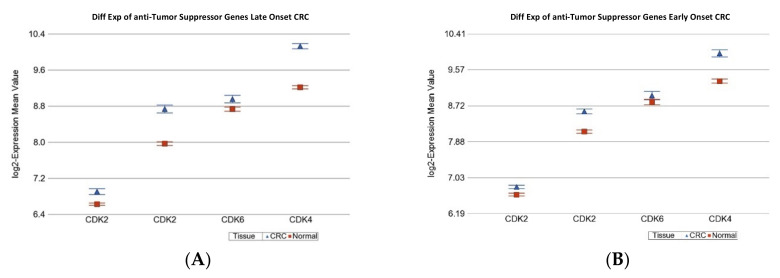
Differential gene expression of anti-TSG genes in paired CRC tissue (in blue) and healthy colonic mucosa (in red). Gene probes are arranged on the *x*-axis by expression level, and the mean of log2 transformed expression value is shown on the *y*-axis. For many genes, there were multiple probes on the chip. Gene symbols for all the gene probes could not be shown on the *x*-axis. Data from patients from the LOCRC group are shown on the left (**A**), and data from patients of the EOCRC group are shown on the right (**B**). The average magnitude of over-expression was significantly higher (*p* = 7.59 × 10^−3^) if the patient had LOCRC [1.46-fold change (95% CI 1.38 to 1.55)] compared to those with EOCRC [1.29-fold change (95% CI 1.20 to 1.39)].

**Figure 3 ijms-23-14261-f003:**
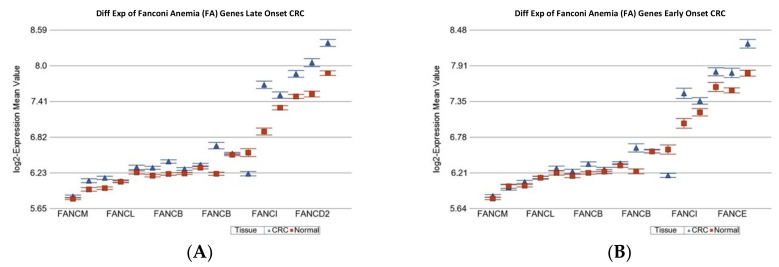
Differential gene expression of Fanconi anemia genes in paired CRC tissue (in blue) and healthy colonic mucosa (in red). Gene probes are arranged on the *x*-axis by expression level, and the mean of log_2_ transformed expression value is shown on the *y*-axis. For many genes, there were multiple probes on the chip. Gene symbols for all of the genes could not be shown on the *x*-axis. Data from patients from the LOCRC group are shown on the left (**A**), and data from patients of the young onset group are shown on the right (**B**). The average magnitude of over-expression was significantly higher (*p* = 4.86 × 10^−4^) if the patient had old onset [1.15-fold change (95% CI 1.13 to 1.17)] compared to those with young onset [1.09-fold change (95% CI 1.06 to 1.12)].

**Figure 4 ijms-23-14261-f004:**
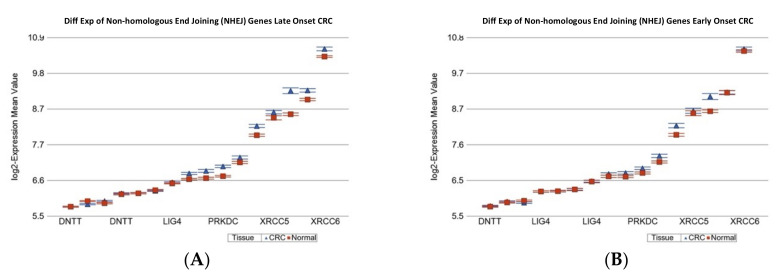
Differential gene expression of non-homologous end joining genes in paired CRC tissue (in blue) and healthy colonic mucosa (in red). Gene probes are arranged on the *x*-axis by expression level, and the mean of log_2_ transformed expression value is shown on the *y*-axis. For many genes, there were multiple probes on the chip. Gene symbols for all of the genes could not be shown on the *x*-axis. Data from patients from the LOCRC group are shown on the left (**A**), and data from patients of the young onset group are shown on the right (**B**). The average magnitude of over-expression was significantly higher (*p* = 6.13 × 10^−4^) if the patient had old onset [1.12-fold change (95% CI 1.10 to 1.14)] compared to those with EOCRC [1.06-fold change (95% CI 1.04 to 1.09)].

**Figure 5 ijms-23-14261-f005:**
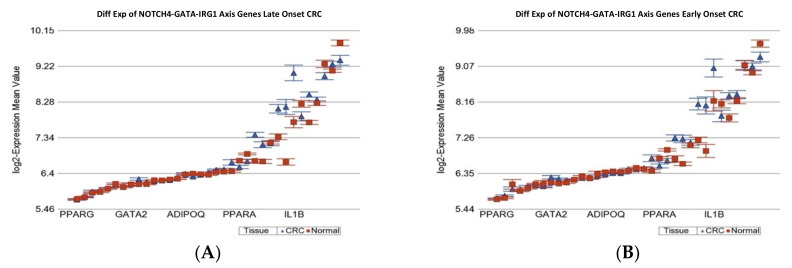
Differential gene expression of NOTCH4_GATA4-IRG1 genes in paired CRC tissue (in blue) and healthy colonic mucosa (in red). Gene probes are arranged on the *x*-axis by expression level, and the mean of log_2_ transformed expression value is shown on the *y*-axis. For many genes, there were multiple probes on the chip. Gene symbols for all the genes could not be shown on the *x*-axis. Data from patients from the LOCRC group are shown on the left (**A**), and data from patients of the young onset group are shown on the right (**B**). The genes were similarly over-expressed in both of the groups.

**Figure 6 ijms-23-14261-f006:**
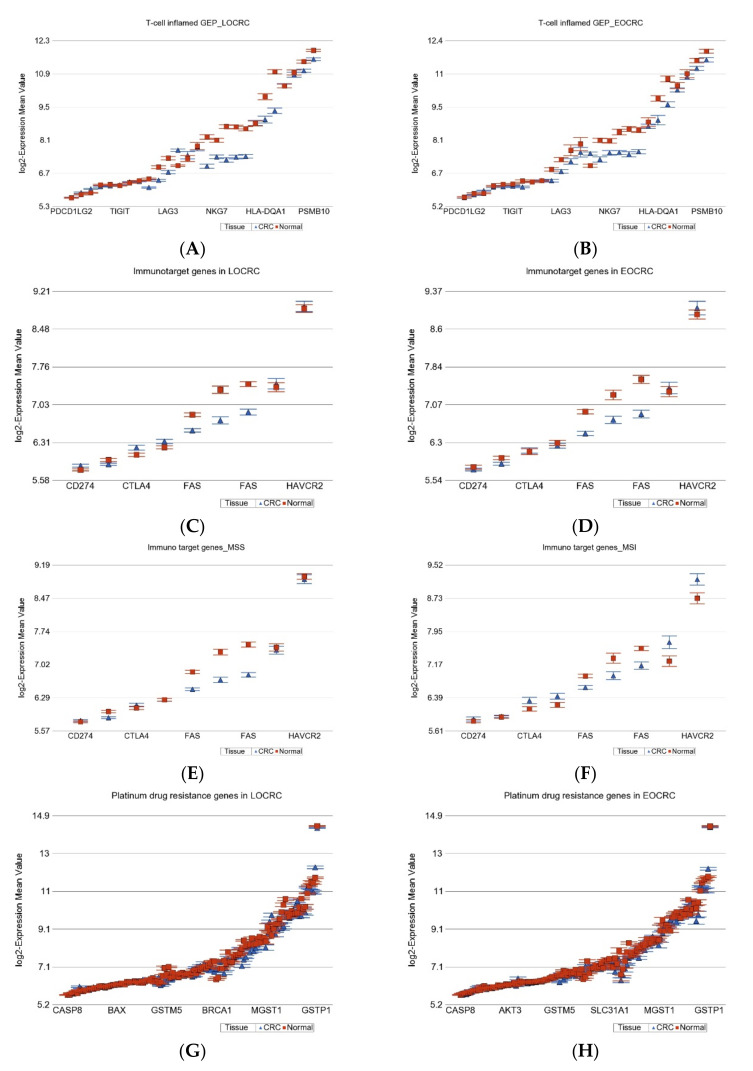
Differential gene expression of few groups of genes related to immune target therapy in paired CRC tissue (in blue) and healthy colonic mucosa (in red). Gene probes are arranged on the *x*-axis by expression level, and the mean of log_2_ transformed expression value is shown on the *y*-axis. For many genes, there were multiple probes on the chip. T-cell inflamed GEP was equally down-regulated in both LOCRC and EOCRC and are shown in (**A**) and (**B**), respectively. Differential expression of few target genes for available check-point inhibitors in LOCRC and EOCRC are shown in (**C**) and (**D**), respectively. Note that the magnitude of difference was similar by age of onset. However, when we divided the patients by MSI status, the difference was seen. Differential expression of the same few target genes in MSS and MSI patients are shown in (**E**) and (**F**), respectively. Over-expression of CTLA4 and HAVCR2 was seen only in the MSI patients, but not in the MSS group. Differential expression of “Platinum drug resistance” genes in LOCRC and EOCRC are shown in (**G**) and (**H**), respectively. These were equally down-regulated in LOCRC and EOCRC and are shown in (**G**) and (**H**), respectively.

**Figure 7 ijms-23-14261-f007:**
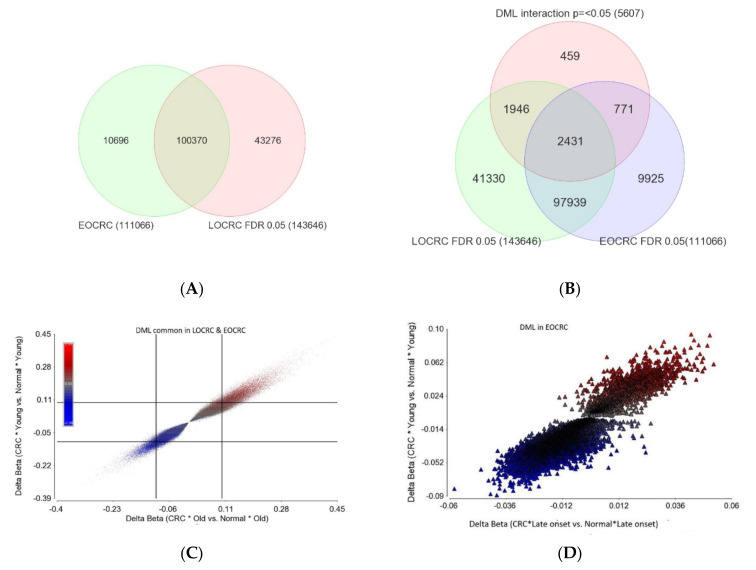
Differential methylation analysis of CRC. The overlap of DML (based on FDR <0.05 only without taking the delta beta into account) in LOCRC and EOCRC is shown in (**A**). In (**B**), we also overlapped the methylation markers that showed different magnitude of delta beta in EOCRC and LOCRC. We identified these markers through the “tissue x age of onset” interaction *p* =< 0.05 in the ANOVA models. (**C**) shows the 97,939 common markers in EOCRC and LOCRC that also had similar magnitude of delta beta in both age groups (because the interaction *p* was >0.05). The scatterplot shows the delta beta of these markers in LOCRC on *x*−axis and the delta beta of these markers in EOCRC on the *y*−axis. (**D**) shows the 9925 DML that were found only in EOCRC, but the delta beta was equally small; I shows the 771 EOCRC specific DML. (**E**) The scatterplot shows the delta beta of these markers in LOCRC on *x*−axis and the delta beta of these markers in EOCRC on the *y*−axis. Although they are greater than LOCRC, the majority of these DML had small delta beta, even in EOCRC. PCA plot using these 771 EOCRC-specific DML is shown in (**F**). Among the three hypomethylated loci with delta Beta =< 0.1, one was *TACC1* gene. The methylation status of *TACC1* gene in LOCRC and EOCRC are shown in (**G**) and (**H**), respectively, showing *TACC1* was hypomethylated only in EOCRC.

**Table 1 ijms-23-14261-t001:** Comparison of histological and molecular changes in LOCRC and EOCRC.

Characteristic	Category	Late-Onset (>40 Years)	Early-Onset (<40 Years)	*Chi-Square Test*
				*p*-Value
Sex	Male	57 (58.2%)	39 (58.2%)	0.995
	Female	41 (41.8%)	28 (41.8%)	
Location	Right Colon	22 (22.4%)	11 (16.4%)	0.327
	Left Colon	20 (20.4%)	10 (14.9%)	
	Rectum	56 (57.1%)	46 (68.7%)	
Stage	Stage-1	30 (30.6%)	7 (10.4%)	0.004 *
	Stage-2	26 (26.5%)	16 (23.9%)	
	Stage-3	42 (42.9%)	44 (65.7%)	
Grade	Low	51 (52%)	27 (40.3%)	0.138
	High	47 (48%)	40 (59.7%)	
Lymph Node	Present	56 (57.1%)	23 (34.3%)	0.004 *
	Absent	42 (42.9%)	44 (65.7%)	
TIL	0	58 (59.2%)	41 (61.2%)	0.796
	1	40 (40.8%)	26 (38.8%)	
Signet Ring	Absent	70 (71.4%)	43 (64.2%)	0.325
	Present	28 (28.6%)	24 (35.8%)	
LV Invasion	Absent	71 (72.4%)	36 (53.7%)	0.013 *
	Present	27 (27.6%)	31 (46.3%)	
PN Invasion	Absent	90 (91.8%)	53 (79.1%)	0.018 *
	Present	8 (8.2%)	14 (20.9%)	
CEA (ng/mL)	Mean	50.221	30.133	0.180 #
	(SD)	100.584	65.732	
Microsatellite	MSI	26 (26.5%)	15 (22.4%)	0.545
	MSS	72 (73.5%)	52 (77.6%)	
KRAS (rs112445441)	Wild	68 (70.8%)	47 (70.1%)	0.925
	Mutant	28 (29.2%)	20 (29.9%	
BRAFV600E	Wild	89 (92.7%)	62 (93.9%)	0.76
	Mutant	7 (7.3%)	4 (6.1%)	
EGFR	Wild	96 (100%)	67 (100%)	NA
	Mutant	0 (0%)	0 (0%)	
Telomere	Absent	28 (30.8%)	23 (37.7%)	0.375
Shortening	Present	63 (69.2%)	38 (62.3%)	

* Significant at <0.05 level; # *t*-test.

**Table 2 ijms-23-14261-t002:** LVI and PNI stratified by stage.

Characteristic	Stage	Status	Late-Onset (>40 Years)	Early-Onset (<40 Years)	*Chi-Square Test* *p*-Value
LV Invasion	Stage-1	LVI Absent	26 (86.7%)	6 (85.7%)	0.947
		LVI Present	4 (13.3%)	1 (14.3%)	
	Stage-2	LVI Absent	23 (88.5%)	9 (56.3%)	0.017 *
		LVI Present	3 (11.5%)	7 (43.8%)	
	Stage-3	LVI Absent	22 (52.4%)	21 (47.7%)	0.666
		LVI Present	20 (47.6%)	23 (52.3%)	
PN Invasion	Stage-1	PNI Absent	29 (96.7%)	7 (100%)	0.624
		PNI Present	1 (3.3%)	0 (0%)	
	Stage-2	PNI Absent	24 (92.3%)	15 (93.8%)	0.860
		PNI Present	2 (7.7%)	1 (6.3%)	
	Stage-3	PNI Absent	37 (88.1%)	31 (70.5%)	0.044 *
		PNI Present	5 (11.9%)	13 (29.5%)	

* Significant at <0.05 level.

**Table 3 ijms-23-14261-t003:** Differential expression of cancer-related pathways in LOCRC and EOCRC.

Gene Set	Interaction *p*	Late-Onset (>40 Years)			Early-Onset (<40 Years)		
		Fold Change	(95% CI)	*p*	Fold Change	(95% CI)	*p*
Growth Factor Receptors	6.64 × 10^−3^	−1.12	(−1.16 to −1.08)	1.21 × 10^−8^	−1.02	(−1.08 to 1.03)	0.34
Anti-TSG	7.59 × 10^−3^	1.46	(1.38 to 1.55)	5.23 × 10^−35^	1.29	(1.20 to 1.39)	3.8 × 10^−11^
DNA Repair	0.06	1.07	(1.05 to 1.08)	2.28 × 10^−19^	1.04	(1.02 to 1.06)	5.4 × 10^−6^
Pro-Apoptosis	0.10	1.03	(1.01 to 1.05)	7.46 × 10^−3^	−1.00	(−1.03 to 1.03)	0.99
Tumor Suppressor Gene	0.10	−1.18	(−1.23 to −1.13)	3.41 × 10^−12^	−1.11	(−1.17 to −1.04)	8.1 × 10^−4^
Hexokinase	0.32	−1.20	(−1.26 to −1.14)	5.64 × 10^−13^	−1.15	(−1.23 to −1.08)	1.6 × 10^−5^
Warburg Effect	0.36	1.32	(1.21 to 1.43)	1.22 × 10^−10^	1.24	(1.11 to 1.38)	1.2 × 10^−4^
Anti-Apoptosis	0.58	−1.10	(−1.15 to −1.05)	1.27 × 10^−4^	−1.07	(−1.14 to −1.01)	0.02
Caspases Initiator	0.69	1.03	(1.01 to 1.05)	1.27 × 10^−3^	1.02	(−1.00 to 1.05)	0.05
p53 suppressor	0.76	1.00	(−1.01 to 1.02)	0.66	−1.00	(−1.02 to 1.02)	9.6 × 10^−1^
Caspases Executor	0.77	−1.18	(−1.23 to −1.14)	4.29 × 10^−18^	−1.19	(−1.25 to −1.14)	2.2 × 10^−12^
Growth Factors	0.78	−1.18	(−1.26 to −1.10)	3.51 × 10^−6^	−1.20	(−1.31 to −1.09)	9.1 × 10^−5^

**Table 4 ijms-23-14261-t004:** Differential expression of DNA damage repair genes in LOCRC and EOCRC.

Gene Set	Interaction *p*	Late-Onset (>40 Years)			Early-Onset (<40 Years)		
		Fold Change	(95% CI)	*p*	Fold Change	(95% CI)	*p*
Mismatch Repair (MMR)	1.16 × 10^−4^	1.12	(1.10 to 1.14)	2.02 × 10^−33^	1.06	(1.03 to 1.08)	7.01 × 10^−6^
Fanconi Anemia (FA)	4.86 × 10^−4^	1.15	(1.13 to 1.17)	1.50 × 10^−46^	1.09	(1.06 to 1.12)	1.42 × 10^−11^
Non-homologous end joining (NHEJ)	6.13 × 10^−4^	1.12	(1.10 to 1.14)	5.60 × 10^−34^	1.06	(1.04 to 1.09)	3.28 × 10^−7^
Microhomology mediated end joining (MMEJ)	9.09 × 10^−4^	1.26	(1.22 to 1.30)	1.29 × 10^−49^	1.16	(1.12 to 1.21)	4.24 × 10^−14^
Translesion Synthesis (TLS)	0.03	1.01	(−1.01 to 1.03)	0.30	−1.02	(−1.05 to 1.00)	0.05
Homologous Recombination (HR)	0.15	1.09	(1.07 to 1.10)	5.92 × 10^−39^	1.07	(1.05 to 1.09)	1.82 × 10^−16^
Nucleotide Excision Repair (NER)	0.19	1.08	(1.07 to 1.09)	1.76 × 10^−41^	1.07	(1.05 to 1.09)	2.56 × 10^−18^
Checkpoint Signaling	0.21	1.08	(1.06 to 1.09)	6.56 × 10^−18^	1.06	(1.03 to 1.08)	4.55 × 10^−7^
Base Excision Repair (BER)	0.40	1.06	(1.04 to 1.07)	4.12 × 10^−15^	1.05	(1.03 to 1.07)	7.80 × 10^−7^
Direct Reversal Repair (DRR)	0.49	1.20	(1.12 to 1.28)	2.96 × 10^−7^	1.15	(1.05 to 1.26)	2.01 × 10^−3^

**Table 5 ijms-23-14261-t005:** Differential expression of replication stress genes in LOCRC and EOCRC.

Gene Set	Interaction *p*	Late-Onset (>40 Years)			Early-Onset (<40 Years)		
		Fold Change	(95% CI)	*p*	Fold Change	(95% CI)	*p*
DNA Replication Repair	2.54 × 10^−8^	1.14	(1.13 to 1.15)	9.40 × 10^−184^	1.09	(1.08 to 1.10)	2.60 × 10^−53^
Metabolism	5.96 × 10^−6^	1.12	(1.10 to 1.14)	1.34 × 10^−24^	1.03	(1.00 to 1.06)	0.03
Cell Movement	6.86 × 10^−4^	−1.06	(−1.07 to −1.05)	3.59 × 10^−26^	−1.03	(−1.04 to −1.01)	1.25 × 10^−4^
RNA Processing	0.11	1.05	(1.03 to 1.06)	6.29 × 10^−13^	1.06	(1.05 to 1.08)	5.88 × 10^−14^
GF Signaling	0.13	−1.04	(−1.06 to −1.02)	8.99 × 10^−5^	−1.01	(−1.04 to 1.01)	0.28
Development Regulation	0.16	1.05	(1.03 to 1.08)	4.04 × 10^−7^	1.03	(1.00 to 1.06)	0.04
Cell Survival	0.27	1.02	(1.01 to 1.03)	3.17 × 10^−4^	1.01	(−1.00 to 1.02)	0.18
Immune Regulation	0.63	−1.02	(−1.03 to −1.01)	6.66 × 10^−3^	−1.02	(−1.04 to −1.01)	7.48 × 10^−3^
Protein Translation	0.65	−1.04	(−1.06 to −1.03)	4.68 × 10^−9^	−1.05	(−1.07 to −1.03)	4.50 × 10^−7^
Stress Responses	0.73	1.04	(1.02 to 1.06)	4.78 × 10^−5^	1.04	(1.01 to 1.06)	7.67 × 10^−3^
Cell Cycle	0.8	1.04	(1.03 to 1.06)	1.08 × 10^−8^	1.04	(1.02 to 1.06)	5.14 × 10^−5^
Angiogenesis	0.91	−1.03	(−1.07 to 1.01)	0.18	−1.03	(−1.08 to 1.03)	0.37
Chromatin TF Transcription	0.93	1.01	(−1.00 to 1.02)	0.21	1.01	(−1.01 to 1.02)	0.40

**Table 6 ijms-23-14261-t006:** Association of differential expression of DNA damage repair gene sets and histological finding of LVI and PNI in CRC tissue.

Stratification	Base Excision Repair Genes		Mismatch Repair Genes		Non-Homologous End-Joining	
	Fold Change	(95% CI)	Fold Change	(95% CI)	Fold Change	(95% CI)
**LV Invasion**						
Absent	1.13	(1.11 to 1.14)	1.18	(1.16 to 1.19)	1.11	(1.09 to 1.13)
Present	1.07	(1.05 to 1.08)	1.12	(1.11 to 1.14)	1.07	(1.05 to 1.09)
** *p* **	1.39 × 10^−8^		1.02 × 10^−5^		1.60 × 10^−3^	
**PN Invasion**						
Absent	1.11	(1.09 to 1.12)	1.16	(1.15 to 1.17)	1.09	(1.08 to 1.11)
Present	1.07	(1.05 to 1.09)	1.12	(1.09 to 1.15)	1.07	(1.04 to 1.10)
** *p* **	2.93 × 10^−3^		6.58 × 10^−3^		0.23	
**Stage**						
Stage-1	1.13	(1.1 to 1.15)	1.18	(1.16 to 1.21)	1.11	(1.08 to 1.15)
Stage-2	1.1	(1.08 to 1.12)	1.15	(1.13 to 1.17)	1.09	(1.06 to 1.11)
Stage-3	1.09	(1.07 to 1.10)	1.14	(1.13 to 1.16)	1.08	(1.06 to 1.10)
** *p* **	0.01		0.04		0.18	

## Data Availability

All supporting data are presented in the tables presented in the main manuscript and as supplemental material.

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
