# Peer review of "A Transcriptome and Methylome Study Comparing Tissues of Early and Late Onset Colorectal Carcinoma"

_ijms, 2022, doi:10.3390/ijms232214261_

Round 1

Reviewer 1 Report

Thank you for giving me the opportunity to review your manuscript entitled
“A multi-omics study comparing tissues of Early and Late Onset Colorectal
Carcinoma”. In this article, the authors carried out a comparative analysis
to understand the pathogenesis of CRC patients which was an imperative
need for treating CRC as it still is the third in cancer-related deaths. The
authors presented the study in a systematic manner and explained the
methods and the results clearly. However, there is a minor revision to work
on the language, with punctuation, grammar, and tense in order to meet the
journal’s requirements.

Reviewer 2 Report

The authors compared tissues of early and late onset colorectal carcinoma. The results are interesting, but the term multi-omics does not reflect the methodology used. Thus, I have some recommendations as follows:

1. The abstract must include conclusion and perspectives. This allows to understand the importance of the findings and future directions.

2. The first sentence of the abstract is the same as the introduction. Avoid repetitions.

3. The figures are of low quality.

4. Some sentences need to be supported by references. For example, lines 358 and 362.

5. Lines 364. The cause is largely unknown. This sentence is confusing. Is the cause of the trend or the etiology of this cancer?

6. Research perspectives should be included in the conclusion.

7. The objective of the present work must be clearly described at the end of the introduction. Authors must understand the purpose of the investigation.

8. The term multi-omics does not seem appropriate in this context. The study does not include metabolomics and proteomics data.

9. Statistical analysis must be included in the tables (1 and 2). Asterisk to represent statistical differences facilitate the understanding of the comparison.

Reviewer 3 Report

Dear Authors, 

please, find my comments on the manuscript below:

1.     Ad. Abstract

“Methylation data also correlated with gene expression” - the sentence is too general. It needs to be clarified.

2.     Ad. 2. Results, lines 81-84

CEA was not listed among the features not related to the age of onset (but it is present in Table 1). On the other hand, EGFR mutation is mentioned, but there was no data about EGFR in Table 1.

In Table 1, the significant association between lymph node status and age of the onset of CRC was presented (p=0.004). However, it was not discussed in the Results section.

3.     Ad. 2.1.1. Age of onset of CRC and Differential Gene Expression of Cancer Related gene sets

Figure 1 is not clear. Does it present the expression level of the selected genes from the GFR set of genes? Why EGFR (left side panel) and EERB2 genes are multiplied in the Figure?

The same comments I have on the next Figures 2, 3, 4, 5, 6 etc.

4.     Ad. 2.1.6. Association of differentially expressed “gene sets” with histology in LOCRC and EOCRC

Does this subsection apply to LO and EO as written in the subtitle? The text of the subsection suggests that only the EOCRC dependencies were analyzed. This should be clarified, also the title of Table 6 should be clarified as to whether it relates to the EOCRC       . 
